# Probabilistic Meta-Learning for Bayesian Optimization

## Abstract

Transfer and meta-learning algorithms leverage evaluations on related tasks in order to significantly speed up learning or optimization on a new problem. For applications that depend on uncertainty estimates, e.g., in Bayesian optimization, recent probabilistic approaches have shown good performance at test time, but either scale poorly with the number of data points or under-perform with little data on the test task. In this paper, we propose a novel approach to probabilistic transfer learning that uses a generative model for the underlying data distribution and simultaneously learns a latent feature distribution to represent unknown task properties. To enable fast and accurate inference at test-time, we introduce a novel meta-loss that structures the latent space to match the prior used for inference. Together, these contributions ensure that our probabilistic model exhibits high sample-efficiency and provides well-calibrated uncertainty estimates. We evaluate the proposed approach and compare its performance to probabilistic models from the literature on a set of Bayesian optimization transfer-learning tasks.

## 1 Introduction

Bayesian optimization (BO) is arguably one of the most proven and widely used blackbox optimization frameworks for expensive functions (Shahriari et al., 2015) with applications that include materials design (Frazier & Wang, 2016), reinforcement learning (Metzen et al., 2015), and automated machine learning (ML) (Hutter et al., 2019). In practical applications, BO is repeatedly used to solve variations of similar tasks. In these cases, the sample efficiency can be further increased by not starting the optimization from scratch, but rather leveraging previous runs to inform and accelerate the latest one.

Several approaches to this emerged under the name of transfer-learning (Weiss et al., 2016) and meta-learning (Vanschoren, 2018). Compared to early work by Swersky et al. (2013); Golovin et al. (2017), recent publications leverage the representative flexibility of neural networks, which allows for more powerful models and impressive results (Gordon et al., 2019; Rusu et al., 2019; Garnelo et al., 2018b;a; Zintgraf et al., 2019). Despite these significant advances, only a small subset of algorithms offers the well-calibrated uncertainty estimates on which BO relies to guide its sampling strategy efficiently. Additionally, BO benefits greatly from a meaningful prior over tasks that quickly converges to the true function to provide the highest sample efficiency. Existing work mostly focuses on deterministic models and, for those providing uncertainty estimates, sample-efficiency at test time is often a challenge.

**Contributions** We set out to close this gap and introduce BAyesian optimization with Neural Networks and Embedding Reasoning (BaNNER), a flexible meta-learning method for BO. We go beyond previous work of Perrone et al. (2018) and introduce a generative regression model explicitly conditioned on a low-dimensional latent representation for the tasks. This allows our model to (i) encode a meaningful prior over tasks and (ii) remain highly sample-efficient, since each new task only requires inference over a low-dimensional latent representation. To ensure robust training of our model, we introduce a novel loss function to regularize the latent distribution and optimize our model's hyper-parameters using the available meta-data. We evaluate BaNNER on a set of synthetic benchmarks and two meta-learning problems and compare with the state-of-the-art in the literature.

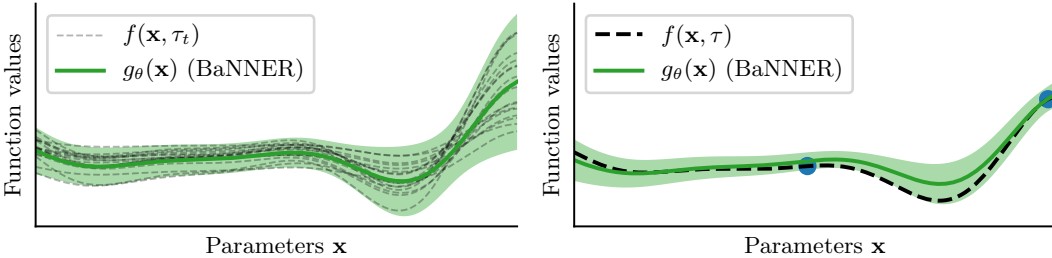

(a) Example tasks and meta-learned prior.

(b) Meta-learned posterior after two observations.

Figure 1: Example application BaNNER. Fig. 1a shows functions $f(\cdot, \tau_t)$ based on samples $\tau_t \sim p(\mathcal{T})$ for a parameterized Forrester function together with the $2\sigma$ confidence interval of the meta-learned prior. In Fig. 1b we see the corresponding posterior distribution after two data points (blue circles) for a specific test-function $f(\mathbf{x}, \tau)$. The confidence intervals contain the true function and collapse quickly, which enables highly-efficient Bayesian optimization. More plots in Fig. 6 (Appendix A.2).

## 2 PROBLEM STATEMENT AND BACKGROUND

Our goal is to efficiently optimize an unknown function $f(\mathbf{x}, \tau)$ over a domain $\mathbf{x} \in \mathcal{X}$ for some unknown but fixed task parameters $\tau$ that are sampled from an unknown distribution $\tau \sim p(\mathcal{T})$. To this end, at each iteration $n$ we can select function parameters $\mathbf{x}_n$ and observe a noisy function value $y_n = f(\mathbf{x}_n, \tau) + \epsilon_n$, with $\epsilon_n$ drawn i.i.d. from some distribution $p_\epsilon$. While our method can handle arbitrary noise distributions, we assume a Gaussian distribution, i.e. $p_\epsilon = \mathcal{N}(0, \sigma^2)$, for the remainder of the paper.

We assume that each evaluation of $f$ is expensive either in terms of monetary cost or time, so that we want to minimize the number of evaluations of $f$ during the optimization process. The most data-efficient class of algorithms for this setting are Bayesian optimization (BO) algorithms, which use the observations collected up to iteration $n$, $\mathcal{D}_n = \{\mathbf{x}_i, y_i\}_{i=1}^{n-1}$, in order to infer a posterior belief over the function values $f(\mathbf{x}, \tau)$. To select parameters $\mathbf{x}_n$ that are informative about the optimum $\max_{\mathbf{x}} f(\mathbf{x}, \tau)$, BO algorithms define an acquisition function $\alpha(\cdot)$ that uses the posterior belief to select parameters as $\mathbf{x}_n = \mathrm{argmax}_{\mathbf{x}} \, \alpha(p(f(\mathbf{x}, \tau) \mid \mathcal{D}_n)$.

While BO algorithms have been studied extensively, their performance crucially depends on the properties of the statistical model used for $f$. The two key requirements for BO algorithms to be data-efficient are *i)*, that the prior belief over $f$ concentrates quickly on the true function $f$ as we observe data in $\mathcal{D}_n$ and *ii)*, that the posterior uncertainty estimates are calibrated, so that the model always considers the true function $f$ to be statistically plausible. The latter requirement means that the true function $f(\cdot, \tau)$ must always be contained in the model's confidence intervals with high probability. Since the task parameters $\tau$ are unknown, in general this requires a conservative model that works well for all possible tasks $\tau$.

We propose to use meta-learning in order to learn an effective prior (Fig. 1a) that can quickly adapt to a new task $\tau$ (Fig. 1b). We are given data from $T$ previous tasks $\tau_t \sim p(\mathcal{T})$ with $N_t$ observations $\mathcal{D}_t^{\mathrm{meta}} = \{(\mathbf{x}_{n,t}, y_{n,t}\}_{n=1}^{N_t}$ each. We show the resulting generative model on the left in Fig. 2. Meta-learning aims to distill the information in $\mathcal{D}^{\mathrm{meta}}$ into a model $g_\theta$ by optimizing the meta-parameters $\theta$. At test time, we then keep these parameters fixed and use the learned model $g_\theta$ to speed up the optimization of the new function $f(\cdot, \tau)$.

## 3 RELATED WORK

There are several approaches to improve the sample efficiency of BO methods based on information from related tasks. We refer to (Vanschoren, 2018) for a broad review of meta-learning in the context of automated machine learning and focus on the most relevant approaches below.

One strategy to improve sample-efficiency is to initialize the BO algorithm with high-quality query points. These initial configurations can be either constructed to be complementary (Feurer et al., 2014; 2015; Lindauer & Hutter, 2018) or learned based on data set features (Kim et al., 2017). An

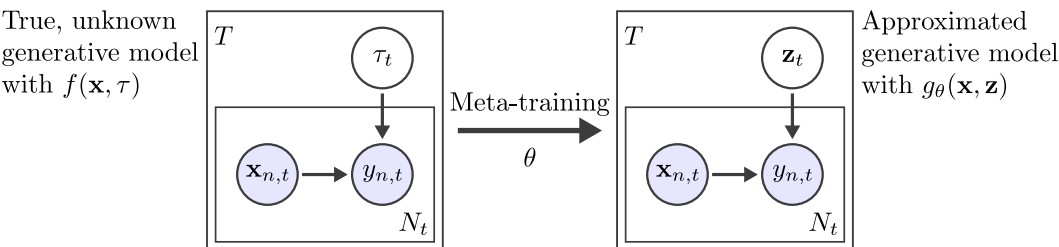

Figure 2: Illustration of the generative models. We approximate the unknown function $f(\mathbf{x}, \tau)$ with $g_\theta(\mathbf{x}, \mathbf{z})$ by meta-learning the parameters $\theta$ based on the noisy observations $y = f(\mathbf{x}, \tau) + \epsilon$. We regularize the model so that we can infer a reliable uncertainty prediction for a new function $f(\mathbf{x}, \tau)$ through approximate inference on $\mathbf{z}$ in our model. Our method results in reliable confidence intervals and is able to adapt quickly, see Fig. 1.

alternative strategy is to transfer knowledge between tasks by adapting the acquisition function. These approaches balance exploration and exploitation by weighting the usefulness of a given $\mathbf{x}$ not only on the current task, but also on the related tasks. This trade-off can either be heuristically motivated (Wistuba et al., 2015; Wistuba et al., 2018), or learned directly (Volpp et al., 2020). All of these approaches change the BO algorithm, but not the underlying probabilistic model.

Most related to our work are methods that focus on alternative ways to model the observations. Several methods build a global Gaussian process (GP) models across all tasks (Swersky et al., 2013; Golovin et al., 2017; Marco et al., 2017; Feurer et al., 2018; Law et al., 2019), which requires approximations due to the cubic scaling of GPs. A method to avoid this scaling is to use a shared neural network to learn specialized features for Bayesian Linear Regression (BLR). The method, dubbed adaptive BLR (ABLR)(Perrone et al., 2018), learns a new BLR model for each task based on the shared neural network features. This allows ABLR to quickly adapt to new tasks and scale better than GPs, but limits its predictive power for a small number of test data points. We consider ABLR to be the state-of-the-art for our setting and use it as a baseline in our experiments.

The problem of adapting a model to a new problem is not unique to BO, but arises in other fields too. Recent progress in transfer learning (Weiss et al., 2016) and meta-learning(Vanschoren, 2018) is concerned with the adaptation of a ML model from one or multiple related tasks to a new one. This transfer often focuses on data-efficiency, which makes them similar to BO. While Frameworks for general (probabilistic) model adaption (Finn et al., 2017; 2018) exist, our contribution relates to a subset of approaches that directly model latent task distributions (Gordon et al., 2019; Rusu et al., 2019; Garnelo et al., 2018b;a; Zintgraf et al., 2019). All of them model the relationship between tasks using the task specific variables, latent variables, and a model that adapts to new tasks by using these latent variables (often as an input). Most of these meta-learning algorithms do not consider active learning as an application and focus on deterministic models $g_\theta$. In the following, we propose a method to meta-learn a probabilistic prior model that we can use together with approximate inference in order to estimate a posterior distribution for BO.

## 4 PROBABILISTIC META-LEARNING FOR BAYESIAN OPTIMIZATION

In this section, we discuss how to approximate $f(\mathbf{x}, \tau)$ with a generative model based on the metadata $\mathcal{D}^{\text{meta}}$ and how to use this model to infer predictions for BO on a new task, given test-data $\mathcal{D}_n$. We use a neural network $g_\theta(\mathbf{x}, \mathbf{z})$ to approximate the unknown function $f(\mathbf{x}, \tau)$. Next to the function inputs $\mathbf{x}$, the network depends on trainable meta-parameters $\theta$ shared between all tasks, and an encoding $\mathbf{z} \in \mathcal{Z} \subseteq \mathbb{R}^d$ of the task-specific parameters $\tau$. We refer to the latter as task embeddings in the following, since they represent unknown task parameters $\tau$. In addition to being used as additional inputs to the network $g$, the latent parameter space $\mathcal{Z}$ can also include task-dependent network parameters, e.g., weights of the output layer. Since we do not know $\tau$ and $p(\mathcal{T})$, we can not match $\mathbf{z}$ to $\tau$ directly. Instead, we assume a fixed and known prior distribution $p(\mathcal{Z})$ to turn $g_\theta$ into a generative network. Without loss of generality, we focus on a Gaussian prior $p(\mathcal{Z}) = \mathcal{N}(\mathbf{0}, \mathbf{I})$ in the following. The resulting approximative generative model is shown on the right in Fig. 2.

**Meta training** The goal of meta-learning is to select appropriate global parameters $\theta$ so that for each task $\tau \in \mathcal{T}$ there exists a corresponding latent encoding $\mathbf{z} \in \mathcal{Z}$ with $f(\mathbf{x}, \tau) = g_\theta(\mathbf{x}, \mathbf{z})$.

Since we have no information about $\tau$, we train a separate task embedding $\mathbf{z}_t$ for each task $\tau_t$ by optimizing them jointly with the parameters $\theta$. We can view the resulting model as a variational autoencoder, where we directly optimize over the outputs $\mathbf{z}_t$ of the encoder, without committing to a specific parametric form. We can optimize the model's predictive performance on the meta-data by maximizing the log-likelihood $\mathcal{L}(\mathbf{x}_{n,t}, y_{n,t}; \theta, \mathbf{z}_t) = \log p_\epsilon(y_{n,t} \mid g_\theta(\mathbf{x}_{n,t}, \mathbf{z}_t))$ of the parameters and observations for each of the $T$ meta-tasks together with the corresponding task embedding:

$$\max_{\theta, \mathbf{z}_1, \ldots, \mathbf{z}_T} \sum_{t=1}^{T} \frac{1}{N_t} \sum_{(\mathbf{x}, y) \in \mathcal{D}_t^{\mathrm{meta}}} \mathcal{L}(\mathbf{x}, y; \theta, \mathbf{z}_t) - \lambda \mathcal{R}(\{\mathbf{z}_1, \ldots, \mathbf{z}_T\} \mid\mid p(\mathcal{Z})), \tag{1}$$

where we rescale the likelihood for each meta-task by $N_t$, the number of data points per task, to account for imbalanced data which skews the loss towards tasks with many evaluations. Without regularization, this model could easily overfit to each task by placing the task embeddings $\mathbf{z}_t$ in disjoint areas of the domain. In that case, samples from the Gaussian prior $p(\mathcal{Z})$ would lead to function samples with low probability-mass under $p(\mathcal{T})$. To avoid this, we introduce a regularization term $\mathcal{R}$ scaled by an appropriate constant $\lambda \in \mathbb{R}_{>0}$ in order to get a more uniform loss approximation over $\mathcal{Z}$. We discuss the specific choice that we use to enable reliable and efficient inference below.

**Inference**  Given the meta-trained generative model, we can make predictions about a new task $f(\cdot, \tau)$. That is, after $n \geq 0$ noisy observations of function values in $\mathcal{D}_n$, we can infer a posterior belief over the task embeddings, $p(\mathbf{z} \mid \mathcal{D}_n, \theta)$. While this is generally intractable analytically, approximate inference methods are generally reliable enough to make this tractable. For example, we use Hamiltonian Monte Carlo (HMC) (Neal, 2011) in our experiments. Given samples from this posterior belief, we obtain a posterior belief over function values $f(\mathbf{x}, \tau)$ from a monte-carlo approximation of

$$p(g_\theta(\mathbf{x}) \mid \mathcal{D}_n, \mathbf{x}, \theta) = \int g_\theta(\mathbf{x}, \mathbf{z}) \, p(\mathbf{z} \mid \mathcal{D}_n, \theta) \, \partial \mathbf{z} = \mathbb{E}_{\mathbf{z} \sim p(\mathbf{z} \mid \mathcal{D}_n, \theta)} [g_\theta(\mathbf{x}, \mathbf{z})]. \tag{2}$$

We can view this posterior inference as a task-specific adaptation of the parameters $\mathbf{z}$ in order to match $g$ to $f$. In contrast to deterministic meta-learning approaches, here the adaptation does not happen directly through an optimization process such as gradient descent, but through approximate Bayesian inference. This enables us to naturally capture uncertainty depending on the amount of data in $\mathcal{D}_n$ that we condition on.

**Regularizing the Latent space**  We can only expect the inference over the predictive distribution in (2) to work well if the two generative models in Fig. 2 encode similar distributions over functions; that is, if $f(\mathbf{x}, \tau)$ with $\tau \sim p(\mathcal{T})$ has a similar distribution as $g_\theta(\mathbf{x}, \mathbf{z})$ with $\mathbf{z} \sim p(\mathcal{Z})$ for all $\mathbf{x} \in \mathcal{X}$. This generally requires each sample $\mathbf{z} \sim p(\mathcal{Z})$ to be associated with a meaningful function $g_\theta(\mathbf{x}, \mathbf{z})$. In this paper, we achieve this through regularizing $\mathbf{z}_t$ to be close to the prior distribution $p(\mathcal{Z})$ and regularizing the network $g_\theta$ to be smooth. Regularizing $p(\mathcal{Z})$ encourages a meaningful prior over functions where each sample has probability mass under $p(\mathcal{T})$, while the regularization of $g$ ensures that samples from the posterior $p(g_\theta(\mathbf{x}) \mid \mathcal{D}_n, \mathbf{x}, \theta)$ vary smoothly across $\mathcal{Z}$.

To regularize the task embeddings one might be tempted to use the log-likelihood of the prior $p(\mathcal{Z})$. However, Tolstikhin et al. (2018) show that this can lead to poor inference results, since a Gaussian prior regularizes $\|\mathbf{z}_t\|_2$, rather than covering the probability mass of $p(\mathcal{Z})$ uniformly. Instead, Ghosh et al. (2020) propose to use a deterministic regularizer during training and use density estimation to determine the prior $p(\mathcal{Z})$.

We propose a novel regularizer that is both deterministic (does not require stochastic approximations) and regularizes the empirical distribution of $\mathbf{z}_t$ to a Gaussian prior $p(\mathcal{Z}) = \mathcal{N}(\mathbf{0}, \mathbf{I})$ in a tractable way. We are inspired by the Kolmogorov-Smirnov test for one-dimensional distributions, which we use to compare the empirical cumulative distribution function (CDF) of the elements $[\mathbf{z}_t]_i$ in $\mathbf{z}_t$ with the marginal CDFs over the dimensions of the prior $p(\mathcal{Z})$. We show an example in Fig. 5 in the appendix. The empirical CDF is defined as $F(z, d) = \frac{1}{T} \sum_{t=1}^{T} \mathbb{I}([\mathbf{z}_t]_d \leq z)$, where $\mathbb{I}([\mathbf{z}_t]_i \leq z)$ is the indicator function that returns one if the $i$th component of $\mathbf{z}_t$ is smaller or equal than $z$ and zero otherwise. In addition to the marginals, we account for correlations by regularizing the empirical covariance matrix $\mathrm{Cov}(\{\mathbf{z}_1, \ldots, \mathbf{z}_T\})$ to be close to that of $p(\mathcal{Z})$, which is the identity matrix in our

---

**Algorithm 1** Probabilistic Meta-learning for Bayesian Optimization

---

1: Given meta-data $\mathcal{D}_t^{\text{meta}}$ for tasks $t = 1, \ldots, T$
2: $\theta \leftarrow$ meta-train by minimizing loss (1) on the metadata
3: New, unknown task $\tau \sim P(\tau)$, $\mathcal{D}_1 \leftarrow \emptyset$
4: **for** iteration $n = 1, \ldots$ **do**
5:     Approximate predictive distribution $p(g_\theta(\mathbf{x}) \mid \mathcal{D}, \mathbf{x}, \theta)$ in (2)
6:     $\mathbf{x}_n = \arg\max \alpha(p(g_\theta(\mathbf{x}) \mid \mathbf{x}, \theta, \mathcal{D}_n))$         ▷ Optimize BO acquisition function
7:     $\mathcal{D}_{n+1} \leftarrow \mathcal{D}_n \cup \{(\mathbf{x}_n, f(\mathbf{x}_n, \tau) + \epsilon_n)\}$

---

setting. The resulting regularizer,

$$\mathcal{R}\big(\{\mathbf{z}_1, \ldots, \mathbf{z}_T\} \mid\mid p(\mathcal{Z})\big) = \underbrace{\sum_{i=1}^{d} (F([\mathbf{z}_t]_i) - \Phi([\mathbf{z}_t]_i))^2}_{\text{Match marginal CDF of } p(\mathcal{Z})} + \underbrace{\lambda_{\text{c}} \|\mathbf{I} - \text{Cov}(\{\mathbf{z}_1, \ldots, \mathbf{z}_T\})\|_{\text{F}}^2}_{\text{Match second moment of } p(\mathcal{Z})}, \quad (3)$$

trades off the loss for the empirical, marginal CDF of the task embeddings relative to the one for the second moment through a scaling parameter $\lambda_{\text{c}}$. Unlike the original Kolmogorov-Smirnov test, (3) uses the average distance between the CDFs at the points $\mathbf{z}_t$. We found this to lead to more stable training than the original formulation which uses the maximum over $\mathcal{Z}$. In practice, many different tests for assessing multivariate normality exist (Korkmax et al., 2014) and could be used instead of (3). For example, Marida's test (Mardia, 1970) also considers higher-order moments. However, we found it to be too computationally expensive relative to the cheap $\mathcal{O}(dT(\log(T) + d))$ complexity of (3) and the latter was sufficient to achieve high performance in our experiments.

While the log-likelihood term $\mathcal{L}$ in (1) encourages a good fit for each meta-task and the regularization $\mathcal{R}$ in (3) forces the meta-task embeddings conform with the prior $p(\mathcal{Z})$, we also have to ensure that $g_\theta$ interpolates between the different task embeddings smoothly in order to obtain meaningful functions $g_\theta(\cdot, \mathbf{z})$ for $\mathbf{z} \sim p(\mathbf{z})$ that are not part of the task-embeddings during training. A simple trick to enforce smoothness is to add noise to the embeddings during training, which is equivalent to regularizing the Hessian $\partial^2 g_\theta(\mathbf{x}, \mathbf{z}) / \partial \mathbf{z} \partial \mathbf{z}$ for Gaussian likelihoods $p_\epsilon$ (Webb, 1994; Bishop, 1995; An, 1996).

**Bayesian optimization** algorithms can directly use the predictive distribution (2) in the acquisition function $\alpha$ in order to select informative parameters $\mathbf{x}_n$. Most acquisition functions only depend on the mean and variance of the predictions or samples from the posterior, all of which can be directly computed from (2). The overall algorithm is summarized in Algorithm 1: We first use the meta-data to minimize the regularized meta-loss in Line 2. In Line 3 we get a new task $\tau$ and start without any test data $\mathcal{D}_1 = \emptyset$. After that, we proceed iteratively and approximate the predictive distribution, select new parameters to use by maximizing BO's acquisition function, evaluate test-task $f(\mathbf{x}_n, \tau)$, and add the data point to our test data set.

**Selecting hyperparameters** Like most machine learning methods, BaNNER depends on hyperparameters that include the regularization constants $\lambda$, $\lambda_{\text{c}}$, parameters of the BO acquisition function $\alpha$, the amount of noise to add for regularization, training-specific hyperparameters like learning-rates and batch-sizes, and inference-specific hyperparameters such as the length and number of the chains that we sample from. In practice, these have to be selected based on the meta-data only, since we typically cannot generate additional meta-training data easily or cheaply. For our experiments, we split the metadata into train and validation sets and select hyperparameters by comparing the average log likelihood of the trained models. For BO tasks, we split each validation task randomly into data points that we condition on in (2) and points that we use to evaluate the likelihood.

To efficiently optimize all these parameters we use BOHB (Falkner et al., 2018), a highly parallel framework to optimize hyperparameters. To speed up computation, we use BOHB's multi-fidelity capabilities and scale the number of meta-training iterations and validation tasks used to compute the validation loss with the fidelity. This allows us to quickly find promising regions of the hyperparameter search space and focus the compute resources there.

## 5 EMPIRICAL EVALUATION

In this section, we evaluate BaNNER in different scenarios and compare it to other methods. We focus specifically on low-dimensional problems where BO outperforms other global optimization methods. As benchmarks we designed five synthetic function ensembles from well-known benchmark function. These include functions where the optimum of tasks varies only locally and benchmarks where the minima vary over the entire domain so that the model must adapt globally. Additionally we evaluate simulated meta-learning tasks, where the objective is to efficiently optimize the hyper-parameters of two machine learning algorithms across different data sets.

We use two variants of BaNNER. One that only considers task embeddings as input and one that additionally considers BLR over the last layer, dubbed BaNNER-BLR in the following. We compare both to several baselines: Random search is a simple yet often surprisingly competitive baseline in the hyperparameter optimization settings (Bergstra & Bengio, 2012; Li et al., 2017). We also consider Gaussian process based BO (GPBO), which is the de-facto default for low-dimensional problems in the BO community. Lastly, we compare to ABLR by Perrone et al. (2018), which we see as the state-of-the-art for our problem setting. We do not consider approaches like stacked GPs (Golovin et al., 2017) and weighted GP ensembles (Feurer et al., 2018) that do not scale to the large meta-data sets we consider.

We implement ABLR and BaNNER in PyTorch (Paszke et al., 2019) and use the open source Gaussian Process implementation from Emukit for GPBO (Paleyes et al., 2019). The latter is a modular BO framework, which we use to evaluate and compare all methods. All experiments use the default BO parameters with expected improvement (EI) as the acquisition function. We will make all model implementations and code to reproduce the experiments presented below openly available after acceptance of the paper.

### 5.1 SYNTHETIC FUNCTION ENSEMBLES

We derive all ensembles of function in this section from well-known functions by replacing parameters of the function with distributions over them. These include one-dimensional quadratic functions and an ensemble of Forrester functions (Forrester et al., 2008), as well as extended multi-dimensional functions Branin, Hartmann3 and Hartmann6 (Dixon & Szego, 1978). We provide details about these functions, their parameter distributions, and the meta data in Appendix B.

The Forrester ensemble shown in our illustrative example in Fig. 1 represents the easiest benchmark. The provided meta data covers both $\mathcal{X}$ and $\mathcal{T}$ well and the optimum often falls into a relatively small region of the search space. As one would expect, the results in the left panel of Fig. 3 show that BO benefits strongly form any meta-learning model. Both variants of BaNNER start with already low regret and keep improving, while ABLR starts consistently worse than random sampling, but improves more quickly than GPBO. The reason for this is that ABLR starts with an uninformed BLR layer that has a symmetric prior over all weights. Since the meta-data is normalized, this means that the mean prediction of ABLR without tests points is equal to the mean of the meta data. Since the weight prior is uniform, the predictive variance is large whenever the neural network features attain large absolute values, which occurs at points where the training data is most extreme. For Forrester, this coincides with the large function values at $x = 1$, which is not close to the optimum.

Our second example consists of one-dimensional quadratic functions. For this benchmark, we provide less meta data per task and allow shifts, so that the minimum of the function can vary over the domain. Methods that quickly estimate the overall global shape of the function can expect small regret. We show results in the right panel of Fig. 3 for 256 different runs. We can see that both variants of BaNNER achieve the lowest mean regret. Surprisingly, ABLR performs worse than random search, indicating that the small number of meta-data points did not allow ABLR to learn representative features, even though the network is capable of approximating the optimal features $1$, $x$, and $x^2$ easily. We suspect that the regularization in BaNNER helped mitigate these problems.

The third ensemble based on the two-dimensional Branin function, has a true latent dimensionality six, $\tau \in \mathbb{R}^6$. The results in the bottom-left panel in Fig. 3 show a similar picture to the two benchmarks above: Both variants of BaNNER perform best from the beginning and GPBO eventually closes the gap. ABLR's initial evaluation are poor, but it quickly improves to outperform Random Search and GPBO for the first 16 iterations, at which point it stagnates without further improvement.

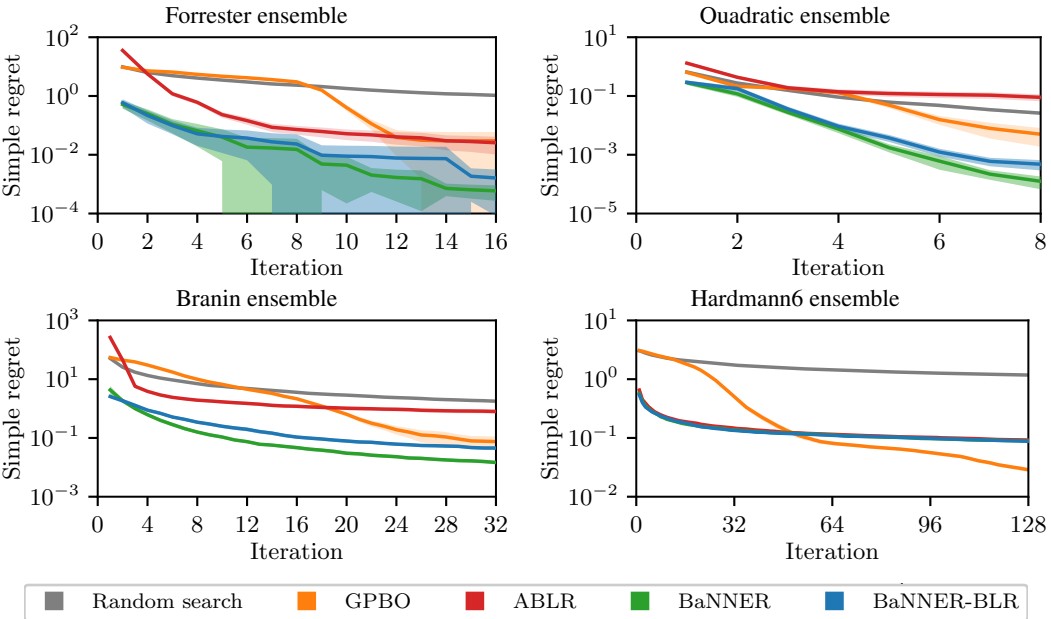

Figure 3: Performance on two synthetic function ensembles. All show the mean $\pm$ the standard error of the mean (95% confidence) for 256 independent evaluations. For the Forrester ensemble, a strong prior without any test data leads to a strong performance right from the start for both variants of our method. For the quadratic functions, the strength of the meta-models lies in knowing the global shape of the functions, rather than knowing a-priory the location of the optimum. ABLR failed to learn useful features from the provided meta-data, possibly due to the over-parametrized network and more broadly distributed $y$ values. With growing complexity and a smaller ratio of number of points per task per dimension, the performance of the meta-learning models degrades and they fail to improve over time, especially for Hartmann6. In all cases, the GPBO method eventually catches up and usually outperforms the other methods, but meta-learning leads to strong early performance in all cases. Additional plots including also an ensemble of Hartmann3 functions can be found in Appendix A.

Our last two synthetic ensembles are based on the Hartmann3 and Hartmann6 functions, where we deliberately deliberately does not contain near-optimal points for meta-training. All meta-learning algorithms perform similarly (see Fig. 3 for Hartmann6 and Appendix A for Hartmann3): After a short improvement, BaNNER and ABLR stagnate and GPBO outperforms them. We attribute this to the growing complexity of the functions in terms of the search space size and the size of the training data set, which does not allow the models to adapt to test points beyond a certain accuracy. For Hartmann6, all three meta-learning algorithms exhibit nearly the same performance, which could indicate that the type of probabilistic model becomes less relevant, if the meta-model lacks the necessary precision around the optimum.

In summary, the synthetic experiments demonstrate the potential improvements of a probabilistic meta-model in BO and also explored its limitations when the meta data does not cover the minima sufficiently. We would like to highlight that in the case of good coverage of both, $\tau$ and $x$, the additional task parameters in the BLR layer of BaNNER-BLR did not provide any benefit.

## 5.2 META-LEARNING SURROGATE BENCHMARKS

We now consider a more practical meta-learning application: tuning the hyperparameters of ML algorithms across different data sets. In this setting, $f(\mathbf{x}, \tau)$ represents the performance of an algorithm on a specific dataset that has unknown properties $\tau$. Instead of optimizing the ML algorithms directly, we use HPOlib2 (Eggensperger et al., 2013), a library dedicated to the evaluation of hyperparameter optimization algorithms. HPOlib2 replaces the costly training step by a cheap lookup based on a large number of hyperparameter evaluations.

We evaluate our method and the baselines on two benchmarks: The training of GLMNET (Genaralized Linear Models with elastic NET regularization) (Friedman et al., 2010) and Ranger, a random forest implementation, (Wright & Ziegler, 2017) and their evaluations on 37 data sets(Kühn et al., 2018b).

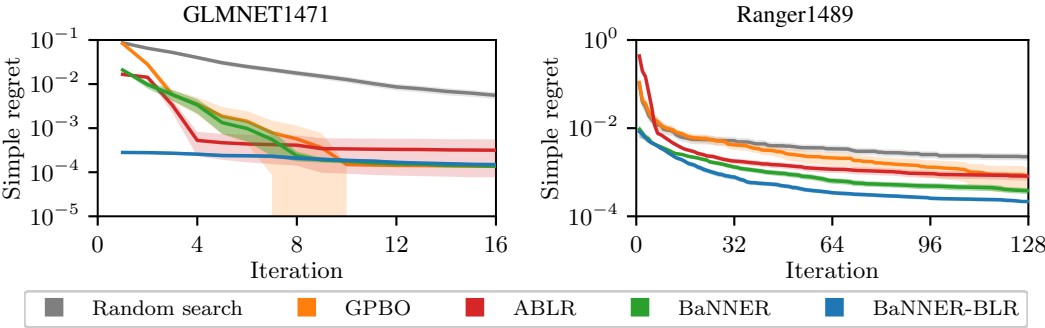

Figure 4: Performance on two surrogate meta-learning problems. Both show the mean $\pm$ the standard error of the mean (95% confidence). We ran every method 256 times with different seeds.

Both of the benchmarks have data sets available on OpenML (Van Rijn et al., 2013) and the collected evaluation data is licensed under creative commons (Kühn et al., 2018). Moreover, the two algorithms are well understood ML methods that usually benefit from optimizing their hyperparameters. We provide more details in Appendix C.

For our experiments, we randomly split the data sets into 32 meta-data sets and 5 test data sets. We refer to these benchmarks as, e.g., GLMNET335 or Ranger1487, where the id corresponds to the data set that the method was trained on. We refer to (Kühn et al., 2018b) details about the data sets and their properties. For the meta training data, we sampled 128, and 512 points for GLMNET and Ranger respectively using the surrogate model. Instead of validating on unseen tasks with unseen hyperparameter settings, we reuse the training tasks, but with an unseen set of $\mathbf{x}$. In this scheme, the hyperparameter optimization tuning BaNNER's parameters cannot estimate the generalization to new tasks. We do this to avoid a computationally expensive cross-validation scheme that would require several fits of the meta-learning models. Our results verfy this simplified validation in our scenario, suggesting that the test tasks are sufficiently similar to the training tasks.

Figure 4 shows representative results on one of data sets per ML method. We see a strong performance on the very first iterations for BaNNER and BaNNER-BLR compared to random sampling. The performance of ABLR's initial guess fluctuates depending on the data set. We attribute this again to the uninformed BLR layer at the beginning, as discussed above. The plots for all data sets can be found in Appendix A. Notably, in this scenario BaNNER-BLR seems to outperform BaNNER. We suspect that the good coverage in $\mathbf{x}$, but the rather poor coverage of $\tau$ (due to the small number of datatsets) could be an explanation. Another difference to the experiments in Section 5.2 lies in the inherent noise of the evaluations, which could also be more effectively countered by adapting the output layer of the network rather than the embedding.

Besides this, the results look similar to the ones above: our method consistently learns a better prior for the first iterations and usually adapts quickly to competitive regret values. Unlike above, GPBO does not perform much better in the shown number of iterations then the other methods. There are a few examples (GLMNET1489 and Ranger1485) where ABLR has the best initial guess and performs the best. We can only speculate for the reasons, but want to point out that 1485 is quite a large data set with many features compared to the meta training data sets, and 1489 is slightly more imbalanced then the average. Data set 1504 turned out to be easy and all methods, even Random Search, perfectly tune both models within a few iterations.

## 6 CONCLUSION

We have presented BaNNER, a novel approach to probabilistic meta-learning by training a generative model that mimics the data generation process. Our meta-loss contains regularization inspired by statistical tests and enables training the model deterministically while still allowing for efficient inference based on multivariate normal prior. In our experiments we use BaNNER as a model in Bayesian optimization (BO) and on meta-learning tasks, which demonstrate higher sample-efficiency than both standard BO with Gaussian processes and ABLR, the state-of-the-art probabilistic model for scalable transfer-learning in BO.

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
