# OpenReview forum: "Probabilistic Meta-Learning for Bayesian Optimization"
_ICLR.cc/2021/Conference — Reject_

### Official Review · AnonReviewer1 · 2020-10-23
**The paper introduces a new approach to accelerate HPO via transfer learning, but a wider range of baselines are needed to demonstrate its benefits.**

**Rating:** 4
**Confidence:** 4

**Review:**

The paper proposes BaNNER, a new transfer learning approach to accelerate hyperparameter optimization based on related tasks. This is based on a generative model learning the data and task distributions. A number of experiments against standard BO and random search, as well as against ABLR, the most closely related transfer learning baseline, indicate that the proposed approach tends to find a good hyperparameter configuration more quickly.

Positives

1. **Significance.** The paper tackles the cold-start problem in BO by leveraging related tasks. This is a highly relevant problem that fits an active line of research. The paper makes a step forward in this direction.

2. **Reproducibility.** The experimental evaluation is very well detailed, with clearly defined error bars and a large number of repetitions (i.e., 256). Implementations of the models and code to reproduce the experiments will also be made evaluable, which will further foster reproducibility.

3. **Clarity**. The paper is generally well-motivated, clear and easy to follow.

Negatives

1. **Missing baselines.** The key concern with the current version of the paper is that more transfer learning baselines would be needed to fully justify the proposed method. Beyond ABLR, a number of additional follow-up and prior works have addressed the transfer learning problem. Many of these are indeed referenced in the related work, such as the initialization strategies to warm-start BO. It is true that these are different as they change the BO algorithm and not the underlying probabilistic model. Nonetheless, this is not a real reason not to compare against at least some of them. Without these comparisons, it is not clear if BaNNER should be the transfer learning method of choice, considering the many alternatives. One could argue that the proposed approach and warm-starting could be seen as complementary, but the authors would need to show in experiments whether there are mutual benefits from combining them or not. The same considerations hold for the following transfer learning HPO methods which prune or construct a search space based on previous related tasks:
[Wistuba, et al.: Hyperparameter search space pruning–a new component for sequential model-based hyperparameter optimization. In Machine Learning and Knowledge Discovery in Databases, 2015]
[Perrone, et al.: Learning Search spaces for Bayesian optimization: another view of hyperparameter transfer learning. In NeurIPS, 2019.  ]

There are also some more recent closely related baselines that are not compared against nor discussed in the related work. For instance, the following work achieves transfer learning while scaling well in the number of hyperparameter evaluations:
[Salinas, et al.: A Quantile-based Approach for Hyperparameter Transfer Learning. In ICML, 2020.]

  Lastly, while it is true that approaches like stacked GPs (Golovin et al., 2017) and weighted GP ensembles (Feurer et al., 2018) do not scale well to large datasets, they scale cubically only in the number of evaluations per task (as opposed to the total number of evaluations across tasks). In the considered experiments this is not necessarily unfeasible. To be comprehensive, one could also subsample the number of data points per task to make these baselines scale and be able to compare against them. It might be the case the using more datapoints will help BaNNER, but this needs to be shown; otherwise, the advantage of scaling to a large number of past evaluations remains theoretical but not demonstrated in practice. Varying the number of evaluations per task could also be seen as a useful ablation study (point 3 below).

2. **Non-challenging problems**. Apart from synthetic examples, the authors consider tuning generalized linear models and random forests with a low-dimensional hyperparameter space. While these are insightful experiments, most approaches for transfer learning in BO are also evaluated on more high-dimensional and ambitious tuning problems, including neural networks tuning/NAS problems.

3. **Not enough ablation studies**. The paper focuses on useful performance comparisons but does not have enough ablation studies. These would give further insights into the proposed method. For instance, it is not clear how performance is affected by changing the number of hyperparameter evaluations available from related tasks. Also, it is not clear how robust BaNNER is to noisy or unrelated tasks. Without these empirical studies, one only gets a shallow understanding of the proposed approach.


Other comments

a. Figure 4 shows some specific datasets per ML method. I appreciate that all results are reported in the appendix. However, ABLR outperforms BaNNER on some datasets and several other datasets lead to very noisy results. To give a sense of overall performance, I would advise the authors to also report global metrics. One possibility is ranking different methods on each dataset and averaging the ranks (to account for heterogeneous loss values).

b. The authors report that ABLR performs best on a few example problems, such as GLMNET1489 and Ranger1485, and conjecture that this is because 1485 is a large dataset with many features and 1489 is more imbalanced. It is not clear to me why dataset dimension and class imbalance should favor ABLR over BaNNER.

c. The method comes with a number of hyper-hyperparameters, which the authors learn on meta-data by running an additional BO loop powered by BOHB. From the perspective of deploying BaNNER, is this not too cumbersome in practice? Do the authors recommend a set of (robust) defaults for their method to be used off-the-shelf or is an additional BO loop always required?

Overall, I vote for rejection for the current version of the paper. The proposed idea is sound, promising and well-motivated, but the method is evaluated against a very limited selection of baselines and it lacks ablation studies. It is not clear with the current experiments how the BaNNER's performance is placed within the context of general transfer learning methods in the literature. I am ready to change my score if the authors compare to a wider range of baselines.

Minor:
1. Some in-line references are not in the correct format (e.g., "We refer to (Vanschoren, 2018)").
2. Typos in this sentence at page 3: "where we deliberately deliberately does not contain near-optimal points for meta-training".

---

> ### Author Response · Authors · 2020-11-18
> **Detailed rebuttal for AnonReviewer1**
>
> Thank you for your detailed review that shows your interest and appreciation for our work. As we mentioned in the general comment, your view that we should evaluate against more baselines is shared by the other reviewers and we aim to include more results in the paper during the next week; including more GP-based benchmarks. Unfortunately we couldn't find an open-source implementation for the method by Salinas, et al. If you know of one please let us know and we'll be able to include results by the end of the discussion period next week.
>
> We appreciate your encouragement to include more ablation studies. As mentioned in the general comment, we are planning an experiment with varying amount of meta-data. Your question how unrelated tasks affect the learning is interesting: Our intuition is that as long as the latent space dimension is large enough relative to the amount of regularization, the model can encode independent tasks in different "corners" of the latent space. Especially together with HPO to select algorithm parameters, that should not be an issue. Given that unrelated tasks are not a key motivation for meta-learning, we will prioritize baseline experiments over this study for the review.
>
> We fully agree that combining our method with others, like warm-starting, is an exciting future direction. However, given that we see the main contribution in a meta-learning method that specifically considers the BO use case that may be beyond the scope of the paper.
>
> a) We will look into summarizing the overall experiments in scalar metrics; we tried to avoid this initially, since these kind of evaluations can often be misleading for comparison, but agree that it could be useful. Reporting average ranks could be an interesting solution that we will investigate.
>
> b) Our conjecture about the performance of ABLR on the mentioned dataset is based on the observation that the ABLR model has the highest uncertainty in regions where the functions usually vary on short length-scales for no or very little data. This is a consequence of the uninformed prior on the BLR weights used. We observed that this uncertainty of ABLR leads to a different exploration for the first iterations compared to BaNNER in the low dimensional examples during early development. This seems to work worse for the synthetic function ensembles where the sampled functions are homogeneous. In turn, this might be an advantage for tasks that are atypical, We conjecture that the data set characteristics of 1485 and 1489 might require different hyperparameter for optimal performance than the bulk of the training data. If time permits, we will try to investigate this in more detail until the end of the discussion period, but we will focus on the additional experiments first.
>
> c) While it is possible to set robust/conservative parameters for some parts of the method (e.g., MCMC sampling), others like the number of latent dimensions and the regularization weight are problem dependent. For the experimental evaluation, we wanted to avoid the risk of us over-optimizing our hyperparameters relative to the baselines. By optimizing the parameters for all methods and baselines we can compare peak-performance, which is a more objective measure. This also aligns well with the original motivation of meta-learning: Invest large-scale compute offline once and then be data-efficient and fast at test-time. We will add a discussion about which parameters are problem-dependent and which are more general to the paper.

---

> > ### Comment · AnonReviewer1 · 2020-11-21
> > **Reference to code and clarification**
> >
> > An open source implementation of Salinas et al is available on GitHub at https://github.com/geoalgo/A-Quantile-based-Approach-for-Hyperparameter-Transfer-Learning.
> >
> > I'd also like to clarify my comment on comparing with warm-start methods. I agree that combining them with the proposed approach is not within the scope of the paper, but this is true as long as some of these methods are included as baselines. I wanted to point out that both these and your approach try to achieve transfer learning for HPO on different levels, so either it should be shown that BaNNER is the better choice or that the two could be combined to further improve performance.
> >
> > In any case, this point falls under my general concern of the paper not including enough baselines. I thank the authors for their work and look forward to the additional experiments comparing to more baselines.

---

### Official Review · AnonReviewer3 · 2020-10-26
**Probabilistic meta-learning paper in "disguise"?**

**Rating:** 4
**Confidence:** 4

**Review:**

SUMMARY OF REVIEW

This paper proposes a probabilistic meta-learning algorithm to establish an informative prior to be used for Bayesian optimization (BO).

The technical contribution lies in the development of the probabilistic meta-learning algorithm instead of innovations made to BO's acquisition function: nearly the entire technical formulation in Section 4 pertains to meta-learning instead of BO. One may argue that this is a probabilistic meta-learning paper in "disguise".

If the aim of this paper is to set up an informative prior, it would be necessary to produce empirical comparisons with the state-of-the-art probabilistic meta-learning algorithms in terms of probabilistic predictions, which is lacking in this paper. In this regard, it would be necessary to perform experiments on higher-dimensional tasks/problems.

One drawback of their proposed work in the context of BO appears to be that unlike some existing meta-BO algorithms, it cannot exploit the GP posterior means and variances when made available from previous BO tasks due to equation 1.

There are a number of major concerns about the technical formulation and the experimental setup and results, as detailed below.


DETAILED COMMENTS

Page 3: The authors say that "Most of these meta-learning algorithms do not consider active learning as an application and focus on deterministic models g_theta." Exactly which probabilistic meta-learning algorithms in the same paragraph as this claim cannot be used for providing a probabilistic prior? Do existing probabilistic meta-learning algorithms not satisfy the requirements of data-efficiency and well-calibrated uncertainty estimates?

Pages 3 & 4: The authors have placed a prior on latent task embedding z but did not seem to consistently apply the Bayesian treatment on z: The task embedding z_t for each task tau_t is optimized using equation 1 without a Bayesian treatment, while the inference or adaptation (equation 2) applies the Bayesian treatment on z. Can the authors discuss and justify this inconsistency? How does it affect the choice of the optimized theta?

Equation 2: If g_theta approximates f, then should the posterior belief of g_theta(x) be an expectation of g_theta(x,z)? Furthermore, g_theta(x) is not defined.

Page 5: The authors say that "we use BOHB's multi-fidelity capabilities and scale the number of meta-training iterations and validation tasks used to compute the validation loss with the fidelity." How is the fidelity exactly determined? Are all the hyperparameters optimized prior to BO? Some hyperparameters need to updated as more BO samples are gathered in every BO iteration. How are these dealt with?

Page 6: The authors say that "We do not consider approaches like stacked GPs (Golovin et al., 2017) and weighted GP ensembles (Feurer et al., 2018) that do not scale to the large meta-data sets we consider." This lack of comparison, especially with the weighted GP ensembles approach, is not well-justified in the context of BO where it is costly to evaluate the unknown objective functions and hence we do not expect large number of samples. The authors are encouraged to perform this empirical comparison.

Page 7: What is the reason for introducing BaNNER-BLR and when would BaNNER-BLR be expected to perform better? The authors have discussed this only briefly in the second last paragraph of Section 5.2.

Figure 3: The fundamental difference between Algorithm 1 vs. GPBO lies in the former exploiting an informative prior based on probabilistic meta-learning (albeit not containing near-optimal points for meta-training) while the latter utilizes an uninformative prior. It is surprising to see that their proposed Algorithm 1 is not being able to undo the effects of the informative prior and hence performing worse than GPBO with an uninformative prior. The authors' justification in the second last paragraph of Section 5.1 was not able to allay my above concern. The use of an informative prior hurts in this case.

Appendix B: The authors seem to have used a large number of meta tasks T=256 and for some synthetic ensembles, a large number of datapoints N_t=1024 per task. In meta-learning, this seems fine. But, in BO tasks where functions are expected to be expensive to evaluate (Section 2), these numbers do not seem practical or reasonable. Can the authors present results with varying, smaller T and N_t?

The authors should consider performing empirical evaluation on hyperparameter optimization of the larger-scale CNNs, which is commonly seen in BO works.


Minor quibbles

Page 2: The *most* data-efficient class of algorithms for this setting are Bayesian optimization (BO) algorithms... Isn't this too strong a claim to make?

Page 3: adaption

Page 6: form

Figure 3: Hardmann, Ouadratic

Page 7: deliberately deliberately

Page 8: verfy

Page 8: datatsets

---

> ### Author Response · Authors · 2020-11-18
> **Detailed rebuttal for AnonReviewer3**
>
> Thank you for your detailed review that shows your interest and appreciation for our work. Besides the general comment, we would like to address your concern in more detail here.
>
> Your assessment of our paper being a "probabilistic meta-learning paper in 'disguise'" is very interesting. Indeed, our contribution lies in the development of a new probabilistic meta-learning method, which (in principle) is applicable to other problems tackled in that field. But besides this very general applicability, we are especially interested in Bayesian Optimization as a method to use probabilistic methods that leverage meta-data in a scalable way. Including other benchmarks for image reconstruction or similar problems seems out-of scope for this paper that tries to investigate meta-learning for Bayesian optimization.
>
> There seems to be a slight misunderstanding about the way we use our model during the optimization. Our model is not an informative prior for a task, but we rather train a parametric model that describes all the available meta-data using a small set of latent variables. These are inferred during the optimization based on the observed data. Our model is not simply a prior for other surrogate models, but tries to represent the variability of the data with latent variables in a generative model. This also relates to your question about Figure 3: Our method is a parametric model of the function, where the parametrization is learned from the meta-data. This means that the function space contained in our parametric model is fixed, while a non-parametric model like a Gaussian Process in principle can fit any data. In that sense, our model cannot "overcome" a certain threshold of accuracy, even for more data due to model mismatch.
>
> We shall revise our claim that "Most of these meta-learning algorithms do not consider active learning as an application and focus on deterministic models $g_\theta$." We tried to say that in other Meta-Learning work, the application of using these models inside an active learning algorithm (like Bayesian Optimization) is simply not studied. The models are evaluated with different criteria and the benchmarks are chosen to cover other relevant topics. We simply wanted to say that we are aiming at a very particular application that has very strong requirements on the uncertainties for regression in the domain of continuous, low-dimensional functions. We did not want to state that other probabilistic Meta-Learning algorithms are not applicable in principle, but rather that their scope differs from our problem.
>
> Our treatment of directly optimizing $z$ was inspired by the work of Gosh et al. "From variational to determin-
> istic autoencoders" cited in the paper. They show that regularization in the latent space can serve as a substitute for the commonly used variational methods (e.g. ELBO loss) for auto-encoders.
> We associate a numerical variable to each of the tasks and optimize that together with the network weights. One could interpret our approach as an auto-encoder where the encoder part is fully non-parametric. Instead of a neural network, the associated latent variables of the tasks are totally independent. Please let us know if that did not address your concern.
>
> Will will try to incorporate your suggestion of studying the same benchmark with varying number of tasks and points per task to study the behavior of the models in different regimes. We have some concerns about your comment of tuning some larger scale CNNs commonly seen in BO works. We have three main issues with this type of benchmarks:
> 1. The tuned models in these benchmarks are rarely state-of-the-art (e.g., a simple CNN on MNIST), which renders the resulting deep learning model not very useful.
> 2. If the models are actually state-of-the-art, the empirical evaluation is so expensive, that no conclusive comparison between different optimization methods can be conducted. Ten independent runs usually leave a lot of uncertainty when comparing different BO methods.
> 3. Since our method requires meta-data, we would have to spend even more computational resources (most of which on random configurations), which is beyond the scope of our work. We see a lot of value in producing such data sets (See e.g. "NAS-Bench-101: Towards Reproducible Neural Architecture Search" by Ying et al.), but there seems no meta-dataset large enough for our evaluation at the moment.
>
> The number of meta tasks and data points per task were chosen according to the problem. We also opted for a uniform distribution of those points to learn a global model. Tackling the question of "How to deal with data that comes from actual BO runs and is non-uniform?" is definitely a very interesting direction to pursue for future research. In the revised version we will aim to provide more experiments with varying amounts of meta-data for a subset of the benchmarks.

---

### Official Review · AnonReviewer4 · 2020-10-28
**Comparisons to standard Multitask BO in a relatively smaller data setting?**

**Rating:** 5
**Confidence:** 4

**Review:**

In this paper, the authors introduce a technique for multitask Bayesian optimization based on meta learning. In general, there are two broad approaches to multi task learning in Bayesian optimization: sharing information between tasks by modelling the correlation between tasks (e.g., Swersky et al 2013), and sharing information between tasks via global weight sharing across over some embedding (e.g., Perrone et al., 2018). This paper falls in to the latter approach, learning a meta learning style task embedding (eqn. 1) to be able to make predictions over a new task given auxiliary task data (eqn. 2).

Overall, my biggest concern is whether the empirical comparison is as well set up as it could be. In particular, this paper compares primarily to ABLR. However, ABLR avoids comparing to standard multitask Bayesian optimization primarily for scalability reasons, using pooled datasets with tens or hundreds of thousands of examples for transfer learning. In this paper, as far as I can tell, each individual task in section 5.2 is limited to 128 and 512 examples respectively. The total dataset size here is significantly smaller, which (1) does not match the setting ABLR was originally used for, and more importantly (2) is well within reach of multitask BO.

First, before jumping in to specific details on this, let me admit outright that some details of the experimental setup in Section 5.2 are simply not clear to me. You say "For the meta training data, we sampled 128, and 512 points for GLMNET and Ranger respectively using the surrogate model." Does this mean you had 32 (number of meta datasets) * (128 data points) =4096 total hyperparameter observations before starting observation, or 128 total? I'm assuming the former? Or did you use the full set of hyperparameter evaluations? I also wasn't clear on what was meant by "Instead of validating on unseen tasks with unseen
hyperparameter settings, we reuse the training tasks, but with an unseen set of x." Does this refer just to the mechanism for setting BaNNER hyperparameters and not to how test tasks / data sets were selected, or were training tasks actually reused as test tasks?

If my assumptions about the empirical set up of the paper are correct (e.g., 4096 and 16384 meta examples), then my biggest issue is whether the playing field that ABLR was compared to was level. Part of the reason ABLR avoids comparing to the "directly learn correlations across tasks" style of BayesOpt that uses multitask kernels is an emphasis on scalability. In their OpenML tasks for warm start on SVM and XGBoost hyperparameter optimization, their meta dataset contains 6.5x10^4 and 5.9x10^5 data points respectively. In this paper, it seems as though the meta datasets are order(s) of magnitude smaller. This may not be the most efficient regime for ABLR, which can rely on larger feature extractors with more total pooled data.

Now, there's an argument to be made for a data efficiency improvement over ABLR (requiring less overall pooled data), but in these settings with on the order of thousands of pooled data points, it becomes trickier in my opinion to justify not comparing to standard multitask Bayesian optimization using multitask Gaussian processes, which works quite well in the data sparse regime and will have no trouble scaling to 32 tasks with 128-512 examples each.

Some of the standard error plotting (Forrester ensemble, GLMNET1471 in the main text, much of figure 8 in the supplementary materials) is clearly broken for some methods. Is this a bug in plotting or actually large standard errors spanning multiple orders of magnitude despite being over 256 trials? Additionally, the y axes on some plots (e.g., Figure 8 bottom row, Figure 9 bottom row) do not have appropriate scales.

---

> ### Author Response · Authors · 2020-11-18
> **Detailed rebuttal for AnonReviewer4**
>
> Thank you for your thorough and detailed review. We share your view that our experimental section is not yet as strong as it could be. As mentioned in the general comment, we are currently working on improving that. We will also further clarify the details of our benchmarks to avoid confusion and misconceptions. Besides these general points, we will try to address your other concerns and questions below:
>
> Our empirical evaluation setup is not quite as different to the ABLR experiments as it might seem. In the ABLR, the first experiment on higher-dimensional quadratic functions has only 29 related tasks with 10 points each (so 290 points in total). The experiments you reference in your review use 30 related tasks with thousands of evaluations each. The SVM problem is a four-dimensional and the XGBOOST problem is a ten-dimensional problem. It is debatable, whether this amount of data is necessary (especially in the SVM problem). The last experiment that Perrone et al. performed has only 5 datasets and the number of total data collected is not mentioned in the text. In contrast, our smallest benchmark "GLMNET" has only 32 training tasks with 128 points in each task (total of 4096 points) and is two dimensional. Our largest benchmark "Hartmann6D" is 6 dimensional with 256 tasks to learn from with 2048 points each (total of 524288 points), which has a comparable number of points to learn from as the largest data set from Perrone et al. with about 590k points. We decided not to repeat their benchmarks, as the data from OpenML is not static and potentially changed since it was carried out in their work. Instead, we opted for synthetic function ensembles to ensure reproducibility and enable us to easily change the number of meta-tasks and points per meta-task and create different test scenarios.
>
> The large variance in some of the plots (the "broken" plots) is the result of a relatively large variance and the logarithmic y axis in the plots. Whenever the mean is negatively impacted by a few outliers, the standard deviation and hence the standard error of the mean can be larger than the mean, which will result in the artifacts you see in these plots. Here an example, say we have $N-1$ runs with regret zero and one with a regret $\Delta$. In this case, the mean is $\Delta/N$, while the standard deviation is $\mathcal{O}(\Delta/\sqrt{N})$. Additionally, the plotting on a logarithmic scale introduces an asymmetry towards larger uncertainty bounds towards the smaller values. On a linear scale, the plot would show the performance going towards zero, with the confidence intervals reaching/including zero. The effect would disappear as the number of repetitions increases.

---

### Official Review · AnonReviewer2 · 2020-10-29
**A probabilistic approach for BO meta-learning**

**Rating:** 5
**Confidence:** 5

**Review:**

In the paper, a probabilistic method is proposed for Bayesian optimization transfer learning (or meta-learning). A latent representation is modeled separately for each task. The surrogate used in BO will be approximated by a neural network with its input being the test point and the latent representation of the task. The meta-training procedure optimizes the prediction loss over the neural network weight and the latent representations of the source tasks. Additional regularizer is added to induce smooth latent representations. For a new task, predictions are made on the new data point using the surrogate model g_\theta(x,z) with a fully Bayesian treatment. The p(z|D,\theta) is approximated by Hamiltonian Monte Carlo. In the experiments, the proposed method is compared to random search, GP-BO, and adaptive Bayesian linear regression.

The latent representation here could be regarded as some model hyperparameter for the neural surrogate, and the p(z|D,\theta) adaptively changes with the including of new data points into the dataset D. The meta-training procedure resembles that of VAE with z being something that can be directly optimized. The experiment shows that the proposed method performs better than the baselines.

My biggest concern is the novelty of the proposed probabilistic approach. It appears that the framework is very similar to that of Neural processes where the latent representation is conditioned on the observations and the prediction is conditioned on the latent representation and the new input data. I think a very clear distinction between this method and Neural processes needs to be made. To me, the only difference is that the method in this paper proposes to directly optimize over z instead of over a variational posterior q(z|D), which is adopted from Ghosh et al. The Neural processes method is directly applicable to the BO problem, contrary to what it is claimed by the authors as “most of these meta-learning algorithms do not consider active learning as an application”. Therefore, NP should serve as the most relevant baseline to compare with in terms of experiments and methodology.

Regarding the experiments, I am curious why the proposed method has a very large variance on the most simple task Forrester. Also, I am not sure why the initial performance of ABLR is even worse than GP-BO, which means meta-learning hurts its performance initially. Some explanation or ablations studies would be helpful here. One motivation of the work is faster transfer learning for BO by using the neural model. However, there is no runtime comparison with the baselines.
Overall, the paper is easy to follow but lacks a direct comparison with NP which most resembles the proposed method. The experiments would be more convincing if more baselines (such as the NPs) could be included.

Typo: Sec 5.1, strongly "form" any meta-learning model --> from

-------------After author's response--------------

As the other reviewers also pointed out, some baselines are missing and ablation studies on meta-data are missing as well. Meanwhile, my concern is that the presented idea in the paper looks very similar to that of the neural process, except the paper is optimizing over a point estimate of z while NP is optimizing over q(z|D). This remains unexplained in the author's response. Therefore I am keeping my original evaluation at the moment.

---

> ### Author Response · Authors · 2020-11-18
> **Detailed rebuttal for AnonReviewer2**
>
> Thank you for your review and the points you raised. We will follow your advice and include the neural processes in our experiments. The regression experiments by Kim et al. in the "Attentive Neural Processes" paper train neural processes to be a replacement for a Gaussian Process, which is used to generate the meta-data by drawing function samples. While the direct application to Bayesian optimization is missing, we agree that the method is directly applicable and will include it in out experiments.
>
> The variance of our method (but also GPBO towards the end) on the Forester method looks surprising, but ultimately stems from the small regret values achieved and a few outliers, where the true optimum is missed. Due to the nature of the ensemble of the Forester functions (see fig. 1a), there are always two local minima (one on the left and one on the right). It is possible that all of the presented optimization algorithms fail and focus on the sub-optimal local optimum. This can be seen in Fig. 7 (in the appendix). When looking at the median and the interquantile range, the values are actually lower than the mean, which indicates that the mean is negatively influenced by a few runs with relatively poor performance.
>
> About the performance of ABLR for the first few iterations: The main reason why ABLR is not performing better in the beginning is rooted in the idea of learning features for a BLR layer with an uninformed prior. Without any data point, the ABLR model will always predict a constant mean of zero regardless of the learned functions. The variance of the predictions is determined by the variance of the feature functions, which results in larger uncertainties in regions where the functions usually vary on a shorter length-scale. In a way, ABLR does not provide a meaningful predictions without data. In the original ABLR paper, they solve this by drawing the first point randomly, which should result in the same performance as random search. Unfortunately, we missed this detail in our first submission and used the predictive mean and variance to sample the first point (which potentially puts ABLR at a disadvantage). We will fix this discrepancy in the revised version of the experiments.

---

### Author Response · Authors · 2020-11-18
**General Rebuttal response**

First, we want to thank all reviewers for the thorough and constructive reviews.
Most pointed to the experimental section as the main weakness, which we will try to address with additional experiments that include more baselines and more scenarios with varying amounts of meta-data.
In particular, we are going to include StackGPs, RGPE, and Neural Processes in our experimental evaluation.
We will try to add additional experiments on one of the synthetic benchmarks and study the performance with a varying amount of meta-data (varying both the number of related tasks and the number of points per task).
We aim to complete these additional experiments by the end of the rebuttal phase, at which point we will update the paper.

---

> ### Author Response · Authors · 2020-11-24
> **Final comment from the authors**
>
> We worked very hard in the past days trying to include as many suggestions from the reviewers as possible into our experiments, including more baselines and a broader variety of benchmarks. While we have preliminary results, we do not feel comfortable updating the paper right now. We want to ensure that our evaluation is correct and comprehensible.
> We want to thank the reviewers again for their constructive and detailed comments that helped improve the paper. We will resubmit an improved version of this work with a stronger empirical evaluation to another venue.

---

### Decision · Program_Chairs · 2021-01-07
**Final Decision**

**Decision:**

Reject

**Comment:**

This paper propose an approach to probabilistic meta-learning for Bayesian optimization. The goal is to accelerate Bayesian optimization under the assumption that multiple related tasks require optimization. In terms of strengths, the paper addresses an important problem as it has applications to efficiently optimizing hyper-parameters over multiple related data sets or models. In terms of weaknesses, the proposed approach is closely related to neural processes, but this connection was not made in the original paper. The authors were unfortunately not able to provide additional insights or results regarding this point during the limited discussion period and as a result, the novelty of the method at the core of this approach is in question. The reviewers also note that the experimental designs and comparisons performed are limited. For some smaller problems, more standard baseline methods like multitask GPs should be applied. The authors have also not compared to a number of other recent methods aimed at scalable transfer learning for hyper-parameter optimization, as detailed in the comments of Reviewer 1. The reviewers agree that the paper is not yet ready for publication.